# Alterations in Brain Activity Induced by Transcranial Magnetic Stimulation and Their Relation to Decision Making

**DOI:** 10.3390/biology12111366

**Published:** 2023-10-25

**Authors:** Lexie Lawson, Stephanie Spivak, Heather Webber, Saeed Yasin, Briana Goncalves, Olivia Tarrio, Sydney Ash, Maria Ferrol, Athenia Ibragimov, Alejandro Gili Olivares, Julian Paul Keenan

**Affiliations:** 1Cognitive Neuroimaging Laboratory, Department of Biology, Montclair State University, 320 Science Hall, Montclair, NJ 07043, USA; lawsonl3@montclair.edu (L.L.); spivaks2@montclair.edu (S.S.); otarrio96@gmail.com (O.T.); ashs1@montclair.edu (S.A.); ferrolm1@montclair.edu (M.F.); ibragimova1@montclair.edu (A.I.); giliolivarea1@montclair.edu (A.G.O.); 2Department of Psychiatry and Behavioral Sciences, The University of Texas Health Science Center at Houston, Houston, TX 77054, USA; heather.e.webber@uth.tmc.edu; 3New York Institute of Technology, New York, NY 10023, USA; saeedysn@gmail.com; 4School of Health and Medical Sciences, Seton Hall University, South Orange, NJ 07079, USA; briegoncalves@gmail.com

**Keywords:** decision-making, inhibitory TMS, choice bias, neural circuits, conscious intention, cognitive-motor functions, reaction times, choice attribution

## Abstract

**Simple Summary:**

In this study, we investigated how inhibiting the motor cortex via Transcranial Magnetic Stimulation (TMS) affects decision-making and awareness of such choices. We employed low-frequency TMS to temporarily inhibit brain activity. During inhibition, participants were asked to select a preference between side-by-side images of everyday items. We discovered that stimulation affected which side people favored when choosing between pictures. The time to make a decision remained the same, as did their explanation for their choices. That is, explanations as to preferred choice almost always had a ‘reason’ behind them. This indicated that choices can be unconsciously influenced below the level of awareness. This study helps us understand the degree to which the unconscious influences our eventual conscious rationalizations.

**Abstract:**

Understanding the intricate dynamics between conscious choice and neural processes is crucial for unraveling the complexity of human decision-making. This study investigates the effects of inhibitory Transcranial Magnetic Stimulation (TMS) on choice bias, shedding light on the malleability of cognitive-motor functions involved in decisions. While reaction times remained unaffected, inhibitory TMS to either the left or right motor cortex led to a significant bias in screen side preference during a choice task. These findings suggest that our cognitive-motor processes underlying decision-making can be unconsciously influenced by TMS. Furthermore, analysis of choice attribution categories revealed individual variability, emphasizing the complex nature of the decision-making process. These insights contribute to the ongoing exploration of the neural mechanisms governing human choice. As the neural basis of free will continues to captivate scientific inquiry, this research advances our understanding of the intricate relationship between neural circuits and conscious intention.

## 1. Introduction

To relate to and exist in the world around us, we must constantly make decisions. While the five second mental debate between soup or salad for lunch, for example, appears to be seamless, it is the direct result of the interplay between neural circuits, cognitive functions, and external stimuli. But are we aware of what choices we make? That is, what hand pressed “↑” to take the elevator to your office this morning? What toe did you tap to the elevator music? Simply put, do we consciously make some decisions but not others? Are some choices free and others predetermined by multiple factors? And if you deliberately made that choice, when did you do so? How long did it take for you to respond, both in terms of your intention and the subsequent execution of the action?

The mechanisms governing human choice and decision-making processes are still not fully understood. In one famous experiment using electroencephalography (EEG) technology, the readiness potential in the brain was observed prior to subjects reporting being aware of their decision to produce movement, such as flexing of the wrist [1]. Following this, numerous researchers suggested there may be a degree of control over our actions, albeit delayed, that suggests awareness of choice follows the choice itself [2,3,4]. Soon et al. (2008) reported that the outcome of a given choice can be encoded in the prefrontal regions of the brain up to 10 s before the participants themselves are aware of that choice. The purpose of this delay is unknown, but in an odd philosophical twist, when the brain activity is measured in real time, the researchers are aware of the decision before the participant [5,6]. Libet’s work is not without controversy, and the results and their interpretation have been highly debated [7].

Critics take issue with Libet’s methods, cognitive experience, and interpretation of the results. In other words, the notion that decisions are made before we are conscious of them is open for debate. It is therefore reasonable that a moderation of claims is prudent, but pursuing the questions raised by Libet could provide insight into human cognition and the degree of freedom in decision-making. Previous studies suggest that the perceived intention behind an action is created in the prefrontal cortex (PFC) and posterior parietal cortex (PPC) [8], while the action itself is created by the motor cortex [9,10]. These findings reveal that the generation of motor circuits may occur before cognition and that much of decision-making is ‘explaining a decision that has already been made’.

Past studies have largely focused on tracking the neural connection between the sensory processing of perceptual stimuli and the resulting choice of category in animals and humans [11,12,13]. Limited studies, however, have investigated the effects of Transcranial Magnetic Stimulation (TMS) on choice bias. TMS is a non-invasive neurophysiological technique that can influence motor circuits [1,14,15]. Researchers can also modulate brain activity with inhibitory transcranial magnetic stimulation [16,17]. Although TMS techniques can make it difficult to pinpoint exact neural circuits, in addition to their effects varying widely across individuals, much can still be investigated.

In one experiment, when the motor area was stimulated with excitatory TMS while a participant was asked to point with either index finger, it resulted in a hand preference that was opposite to the side of stimulation [18]. Another study determined that single-pulse TMS to the left posterior parietal cortex, but not the right, also leads to hand selection bias when performing speed-reaching tests [19]. Last, another experiment found that TMS to the presupplementary motor area of the brain following a spontaneous action shifted both perceived intention and action execution reaction times [5]. These findings suggest that the balance between neural determinism and the conscious experience of free will may play a crucial role in perceptual decision-making.

Building on these studies exploring the targeted effects of TMS, we aim to further understand the mechanisms underlying our conscious experience and decision-making processes. Here, inhibitory TMS of 0.75 Hz was employed to investigate the causal relationship between the motor cortex and hand selection (screen side response) when presented with two side-by-side choices. More specifically, to test whether inhibitory TMS can unconsciously influence choices, participants were presented with images of mundane objects (e.g., paperclips, lint, binder clips) while experiencing sham, left motor cortex (MC), or right motor cortex TMS. Following TMS, we predicted that participants would be biased to one side or another depending on the region stimulated; left motor cortex stimulation is expected to cause more left-screen or hand-side responses, and vice versa. We further predicted that participants would ‘fill in’ the reason why they made their choices. Overall, we hypothesized that inhibitory TMS can be used to unconsciously influence the cognitive-motor aspects of decision-making and one’s attribution of choice.

## 2. Materials and Methods

### 2.1. Participants

Eleven female and six male participants (N = 17) were recruited for this pilot study via flyers posted on campus (aged 18–27). One participant was left-handed, and sixteen were right-handed. Subjects were paid USD 25 to participate. As specified in Montclair State University’s Institutional Review Board’s (IRB MSU 424) policies and the APA guidelines, all participants were treated ethically, and all gave informed consent following appropriate screening [15].

### 2.2. Materials

For the repetitive TMS (rTMS) stimulation, we used a Magstim 200 with a 7 cm coil in the shape of the number eight. A 14” Lenovo monitor was used to present visual stimuli using the Testable software. Lycra swim caps were also worn by the participants for the rTMS administration.

### 2.3. Stimuli

Participants were presented with 140 instances of a choice in which they were asked to randomly choose between two almost identical images, side by side, of various mundane items such as markers, paper clips, pens, etc. (Figure 1). The images depicted the same item with slight variations in properties such as angle, color, or size. The similar and neutral nature of the images creates a 50/50 possibility of either one getting chosen. In addition to participants being asked to randomly choose which one they preferred, they were given five seconds to elaborate on why they chose the image they did, with emphasis on the category of the attribution (“Color, Size, Shape, Texture, and Not Sure”). Stimuli were always matched to each other, and each pair was presented randomly.

### 2.4. Procedure

Participants thoroughly read and signed the informed consent form prior to the start of the experiment. Protocols surrounding COVID-19 were strictly adhered to, such as social distancing, the sanitization of all areas, and the use of protective equipment such as face masks, shields, gloves, and gowns. Participants wore earplugs and tight Lycra swim caps. We utilized the 10–20 system to measure and mark the appropriate brain areas on the worn swim cap, and the proper limits of stimulation were determined. After stimulation, participants were presented with the testable stimuli on the Lenovo monitor and were asked to select a key on the left side of the keyboard with their left hand, choosing the image on the left, or likewise, to indicate the image on the right, using their right hand to select a key on the right side of the keyboard (Figure 2), followed by an explanation for said choice. For all trials, subjects were not informed if the TMS condition employed on either the left or right MC was inhibitory or sham, nor were they informed that images were both repeated and displayed on both screen sides. By doing so, we aim to investigate if we can alter an individual’s perception of free will by disrupting neurons in the left or right motor cortex and inhibiting the motor function of the opposite hand, thus affecting reaction time, screen-side response/preference, and decision-making as a result.

### 2.5. Motor Threshold

Participants were seated with an extended left hand as suprathreshold TMS pulses were administered to determine at which location the greatest motor-evoked potential (MEP) response was given to the contralateral Abductor Pollicis Brevis (APB) muscle. The coil was moved across the scalp until the most responsive region was located. The coil was held at approximately 45 degrees from the hemispheric line. The minimum TMS intensity (MT) was determined so that we reached the threshold when 50% of the TMS pulses created a MEP of >50 mV, as outlined in the procedures by the International Federation of Clinical Neurophysiology (IFCN). During the experiment, active stimulation was delivered at 90% MT.

### 2.6. Experiment

Following the establishment of the MT, a single-pulse TMS was administered to either the right or left motor cortex for five minutes by the principal investigator (PI). After this, participants were presented with 70 stimuli consisting of two similar pictures and instructed to choose between the two and then explain why (Table 1). With either the left or right hand, participants pressed a key on the keyboard on the respective side, selecting either the image on the left with their left hand or the image on the right with their right hand. Participants were given five seconds after each choice to justify their selection. The control condition used sham TMS, in which the TMS coil was held at a 90 degree angle over CZ, aiming to the right or to the left (respectively), such that no TMS was delivered.

## 3. Results

The average reaction time was 472.34 milliseconds (SE = 40.48). A repeated measures ANOVA revealed a non-significant difference across reaction times over the four TMS conditions (F (3,64) = 0.58, *p* = 0.63; sham left M = 505.55, SD = 168.21; sham right M = 473.12, SD = 148.12; inhibitory right M = 450.93.74, SD =163.68; and inhibitory left 459.74, SD = 175.89; Figure 3). To determine if reaction time differed for the left screen side response (M = 474.40, SE = 22.56) vs. the right screen side response (M = 470.15, SE = 23.11), we pooled all responses. It was found that there was no significant difference in reaction times across selected item locations (t (2378) = 0.13, *p* = 0.90, d = 0.005).

To determine if TMS influenced screen-side responses, we first looked at overall responses regardless of TMS condition and found there was none (left: N = 1227). Right: N = 1153; X^2^ (1) = 2.30, *p* = 0.13). We then looked at the screen side response for the control conditions, finding that sham left motor cortex TMS led to 305 right screen and 290 left screen selections. This difference was found to be non-significant (X^2^ (1) = 0.38, *p* = 0.54). For sham right motor cortex TMS, the right screen was selected 277 times and the left screen was selected 318 times. This difference was also non-significant (X^2^ (1) = 2.83, *p* = 0.09; Figure 4).

The test of the first main hypothesis would be that 0.75 inhibitory TMS would influence choice. It was found that inhibition of the left motor cortex resulted in significantly more left screen choices (356), as compared to right screen choices (239; X^2^ (1) = 23.01, *p* < 0.0001). Likewise, it was found that inhibition of the right motor cortex resulted in significantly more right screen choices (347) than left screen choices (248; X^2^ (1) = 16.47, *p* < 0001). These data support the notion that the cognitive-motor aspects of decision-making can be influenced via TMS. As a control, we compared the two sham conditions in terms of screen-side responses. It was found that for right sham, there were 277 right screen choices and 318 left screen choices. For left sham, there were 305 right screen choices and 290 left screen choices. This difference was found to be non-significant (X^2^ (1) = 5.27, *p* = 0.022; Bonferroni alpha = 0.004).

Testing the attribution of choice was the second central examination. Before testing this directly, we first looked at several ancillary details as the participant’s responses to prompts were examined. Over the course of examining the responses, five categories of attribution arose in addition to ‘Not Sure’ (which was what we termed any description of no attribution). The main categories that were finalized were “Color, Size, Shape, Texture, Feeling, and Not Sure”. All responses included at least one of these; several included multiples (Figure 5). Using these six descriptors, we performed a 4 (TMS conditions) × 6 (descriptors) contingency chi-analysis and found no significant difference (X^2^ (15) = 4.75, *p* = 0.099). Color (X^2^ (1) = 0.32; *p* = 0.95), Size (X^2^ (1) = 1.27; *p* = 0.74), Shape (X^2^ (1) = 1.18; *p* = 0.96), Texture (X^2^ (1) = 0.20; *p* = 0.98), Feeling, (X^2^ (1) = 0.47; *p* = 0.92), and critically, Not Sure (X^2^ (1) = 1.52; *p* = 0.68), did not differ across TMS categories (Figure 6). We were mainly interested in what a person attributed, rather than the attribution itself.

We then analyzed the data on an individual item level. It was found that participants switched preferences about 50% of the time (605/1190). That is, they choose a version of the object when presented, following TMS. However, there were two ways to switch following TMS: in the predicted direction or the non-predicted direction (i.e., left TMS should turn a right preference to a left preference). For left TMS, it was found that there were significantly more predicted changes (M = 10.35, SD = 2.26) than opposite changes (M = 7.35, SD = 2.01; t (16) = 4.21, *p* < 0.0006, d = 2.11). The right TMS condition was also significant (t (16) = 2.35, *p* = 0.03, d = 1.18) with the predicted change (M = 10.12, SD = 2.32) being higher than the non-predicted change (M = 7.76, SD = 2.49).

## 4. Discussion

The use of TMS on choice bias has highlighted the neural connection between sensory processing of perceptual stimuli and the resulting choice of category; hand selection in both a finger-pointing and speed-reaching task is contralateral to the side of motor area excitatory TMS stimulation. Reaction times have also been measured using TMS, finding that stimulation shifts both perceived intention and action execution times. However, it is uncertain whether inhibitory TMS has similar effects when testing choice bias. We therefore applied TMS to the motor cortex to understand inhibitory TMS’s influence on both reaction time and choice attribution in a decision-making assignment. By targeting these regions, we aimed to disrupt the neural communication patterns underlying the selection of hand preference and the subsequent attribution of choice. The motor cortex is intricately connected with higher-order cognitive processes, including the prefrontal cortex and posterior parietal cortex, which are known to play essential roles in the intention and execution of actions [8,9,10].

Results show that reaction times between left and right-hand selections were not significantly affected between treatments, indicating that motor response speed may not be substantially impacted by inhibitory TMS on either the left or right motor cortex. We think SMA or frontopolar TMS would result in RT differences, especially if a more ruminative design were employed. Based on work from Bode’s group, SMA and fronto regions are involved for seconds prior to motor action, and TMS in these regions should influence RT.

When exploring how choice attribution is affected by inhibitory TMS, it was found that inhibition of the left motor cortex caused a significantly higher selection of left-screen side responses compared to right-screen side responses. Likewise, right motor cortex inhibition resulted in a significantly higher selection of right screen selections compared to left. This indicates that our cognitive-motor functions involved in decision-making are influenced by inhibitory TMS, thus leading to a bias in screen-side preference. We have found previously [16] that motor cortex stimulation can have a profound influence on cognitive decision-making, and there (as here), we suggest that feedback plays a role in influencing SMA and frontal activations. TMS delivered to motor regions is specific, and cognitive feedback is process-dependent. For example, disrupting the hand region only influences cognitive tasks that involve the hand, while disrupting the motor cortex regions responsible for foot movements only influences tasks that involve the foot [16].

Participants changed their preferences approximately 50% of the time following TMS treatment at the individual item level; most of these switches occurred in the predicted direction (i.e., left inhibitory TMS leading to a left preference or right inhibitory TMS leading to a right preference). Overall, this suggests that an individual’s choice can be influenced by inhibitory TMS applied to the motor cortex, supporting the hypothesis that TMS can unconsciously influence our discussion-making functions. Previous research findings also support this notion, finding that inhibitory TMS to the prefrontal cortex decreased the subject’s preference for risky choices and that transcranial direct current stimulation (tDCS) modulating cortical activity of the left dorsolateral prefrontal cortex (DL-PFC) can influence decision-making processes [20,21].

While these findings contribute to our understanding of the mechanisms underlying decision-making and inhibitory TMS’ role in modulating choice behavior, there are still limitations to the present study. Having a larger sample size would increase our experiment’s generalizability; future studies with larger sample sizes and an additional control condition may help further our understanding of the influence of TMS on choice. Further, the absence of a control group receiving no TMS treatment could be a potential confounding factor. Another consideration is the inherent variability in individual responses to TMS, which can be influenced by factors such as anatomical differences, cortical excitability, and baseline neural activity. Further, because the sound of the TMS was lateralized, this may have had an influence on choice. Though we deem this unlikely, it cannot be ruled out.

As with all TMS studies, brain regions are chosen, and the study is focused on ROIs (regions of interest). As has been noted [14], this method is effective but limited. We have now found encouraging results in one (lateralized) region and now encourage further regions to be examined. Work from Bode and colleagues [11] would suggest fronto-polar targets, and Haggard’s work would suggest SMA targets [9].

We believe that decision-making is a relatively arbitrary process, and as others have shown, there is much back-projecting. This study does not test that directly but does provide supportive evidence. The ‘filling in’ was expected; that is, the fact that TMS caused changes in decisions seemingly without conscious awareness was predicted. We have concluded that this processing is typical and seen across many phenomena [22] and is a core component of the self. Clearly, this speculation is just that, and further studies need to follow up on the current suggestive findings.

If Libet was correct [4,23] in his assertions about the origins of what he termed free will, activation of motor regions and pre-frontal regions should have an influence on choice. We found that here statistically, but not in all cases. The simple explanation would be that TMS did not reach the level needed in all cases, that the participants were motivated to balance out their selections, or that some other unknown process was at work.

All in all, our findings highlight the importance of further studying the mechanisms governing human choice and decision-making. Future research into understanding the neural basis of free will can aid in our approach to other applications, including our ability to resist external factors when making decisions [24,25,26]. Specifically, the present study, exploring how free will and our unconscious beliefs can be manipulated, has implications for ethics, law, addiction, and other mental health and psychiatric disorders treatments. Using TMS to either stimulate or inhibit specific areas of the brain can allow us to study its contribution to the intention and the resulting execution of our actions [2,27].

## 5. Conclusions

This study delved into the intricate interplay between conscious choice and neural processes, emphasizing the effects of inhibitory TMS on decision-making. While our findings revealed no significant impact on reaction times, they unveiled a compelling influence of TMS on choice bias, underscoring the malleability of cognitive-motor functions involved in decision-making. This work contributes to the ongoing exploration of the mechanisms governing human choice and the extent of our conscious control over decisions. By revealing the potential to unconsciously modulate choice behavior through TMS, our study paves the way for broader implications, ranging from ethical considerations to practical applications in law, addiction, and mental health treatments. As the neural basis of free will continues to captivate scientific inquiry, this research advances our understanding of the intricate relationship between neural circuits and conscious intention, inviting further exploration into the depths of human decision-making and its implications for various facets of society and human behavior.

## Figures and Tables

**Figure 1 biology-12-01366-f001:**
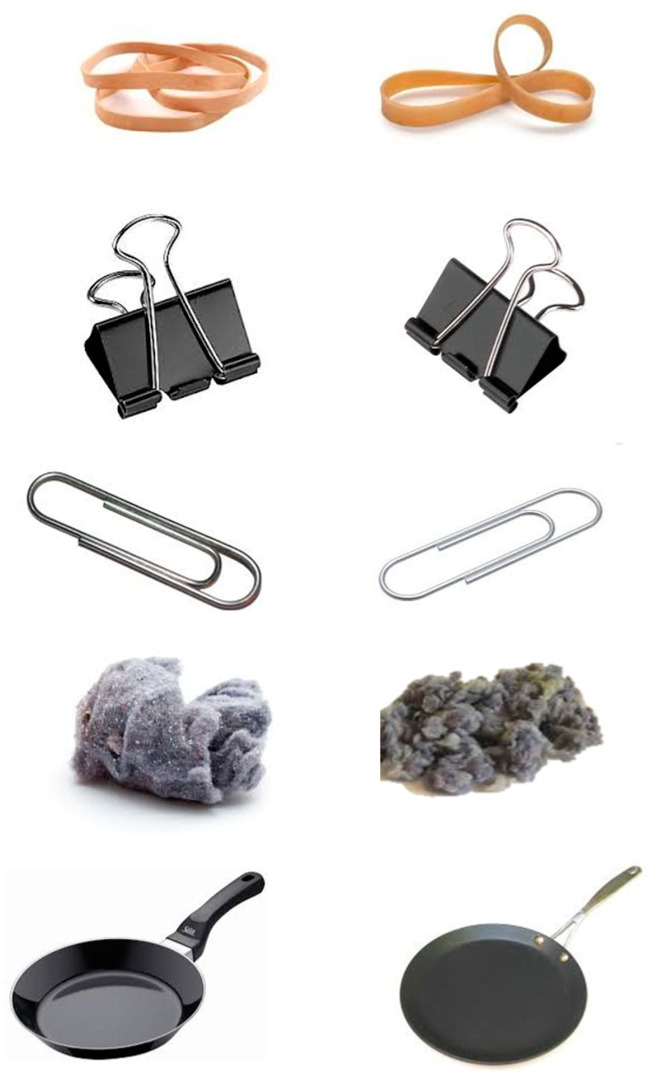
Examples of almost identical visual stimuli displayed side by side. Participants would choose one of the two mundane objects and then explain their choice.

**Figure 2 biology-12-01366-f002:**
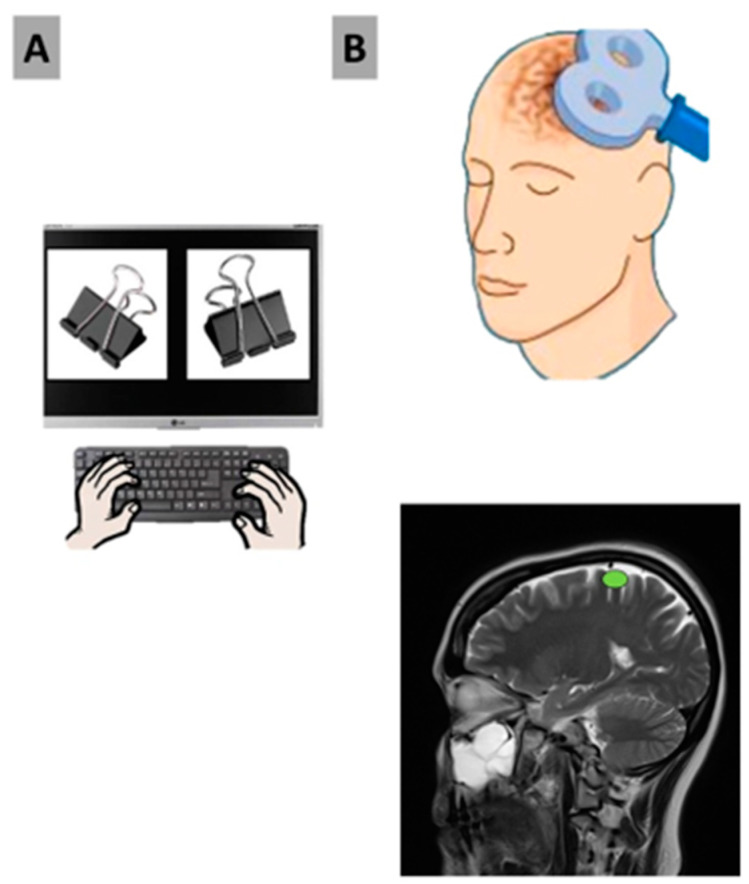
(**A**) Depicts an example of how visual stimuli were presented to participants, with the right hand selecting the right image on the keyboard or the left hand selecting the left image. (**B**) Depicts the TMS coil used to administer a single pulse of TMS to either the right or left motor cortex. CZ (green) traced to A1/A2 served is the initial search points for the hand area.

**Figure 3 biology-12-01366-f003:**
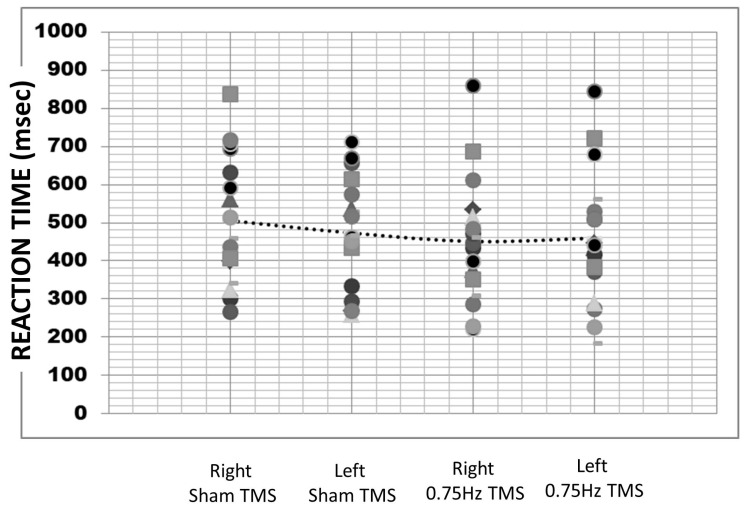
Average reaction times in milliseconds for all 17 participants across the four TMS treatments.

**Figure 4 biology-12-01366-f004:**
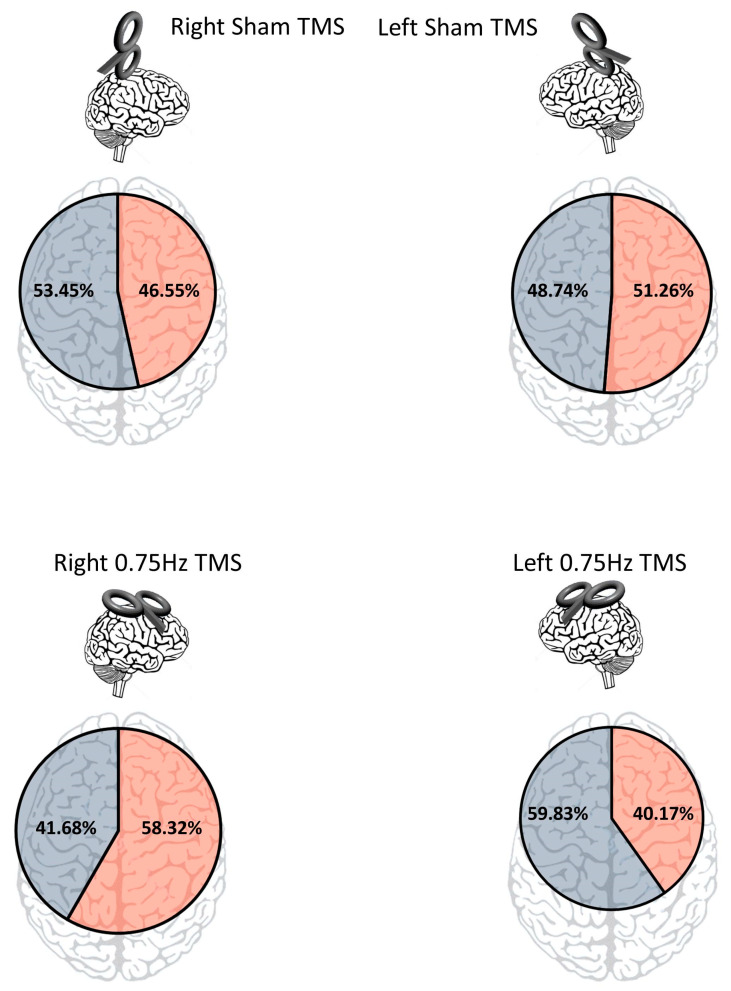
Percentage of right versus left side screen responses across all trials based on each TMS treatment.

**Figure 5 biology-12-01366-f005:**
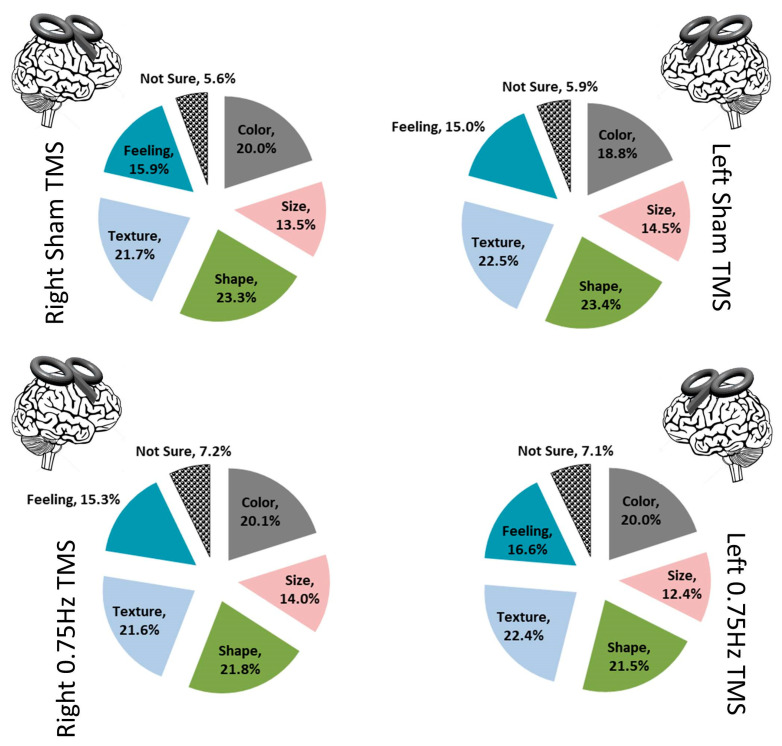
Attribution provided by participants for their choices across all trials based on each TMS treatment.

**Figure 6 biology-12-01366-f006:**
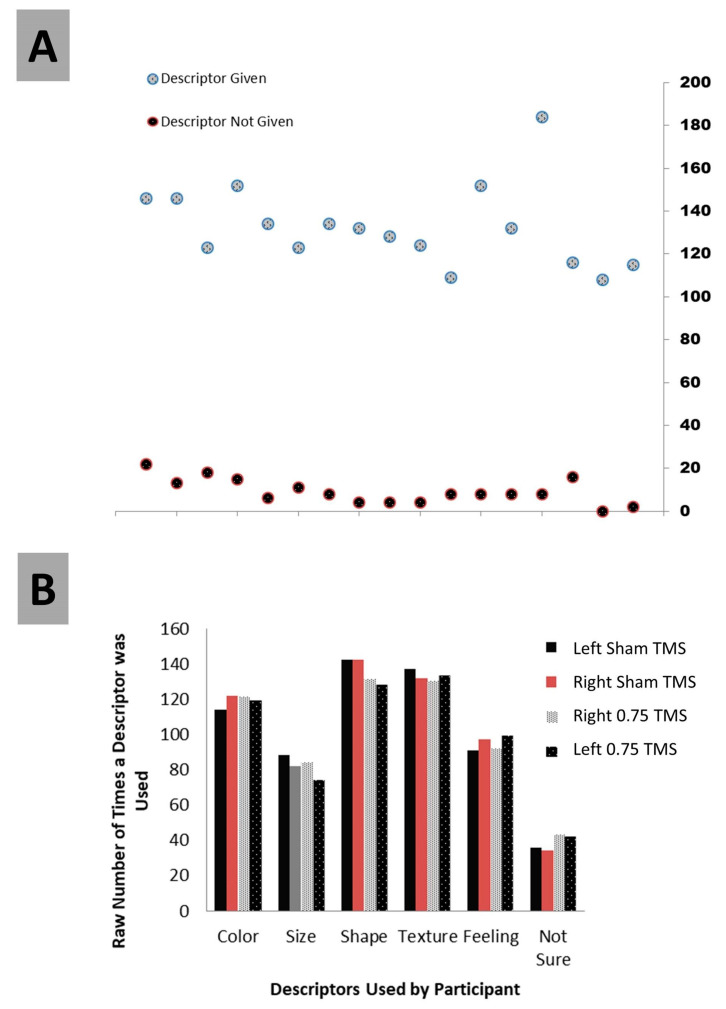
(**A**) Depicts whether participants gave a description (attribution) for each of their choices across all trials or not. (**B**) Depicts what attribution participants described for all trial choices separated by TMS treatments. Note that more than one descriptor was often used for a single stimulus.

**Table 1 biology-12-01366-t001:** Sample descriptors used by participants.

Attribute	Sample Statement
Color	“I chose this one because it was blacker”
Size	“The one of the right seemed bigger”
Shape	“The can seemed rounder”
Texture	“…Lint seemed smoother”
Feeling	“Those pencils made me feel worse”
Not Sure	“I couldn’t tell you”

## Data Availability

Portions of the data presented in this study are available on request from the corresponding author. The data are not publicly available due to IRB restrictions.

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
