# Peer review of "Alterations in Brain Activity Induced by Transcranial Magnetic Stimulation and Their Relation to Decision Making"

_biology, 2023, doi:10.3390/biology12111366_

Round 1

Reviewer 1 Report

This is a very intriguing study on the effects of low 0.75 Hz rTMS to the left or right M1 on left/right stimulus choices (made with the left or right hand). The results show that left M1 inhibition biases people to choose the left side image with the left hand (and vice versa for right M1 inhibition). Furthermore, people seem to rationalize their choices, indicating they are not really aware of what drives them.

I have a number of comments and questions about the paper. Overall, there needs to be clarification on key aspects of the methodology.

---

Introduction. The predictions should spell out the direction of effects: left M1 stimulation is expected to …

Stimuli. There is mention of 140 instances of a choice. Does this mean that there were 140 distinct pairs of picture people could choose from? Or were some pictures/pairs repeated? Why does the Experiment section say that “participants were presented with 60 stimuli consisting of two similar pictures and instructed to choose between the two and then explain why” (Line 154)? Weren’t participants presented with 140 stimuli?

Experiment. Given that Sham was at Cz whereas stimulation of the motor cortex was lateralized, the coil sound was also lateralized. For the sham it is said that the coil was held at 90 decgrees over Cz. What does that mean? That the edge of the coil was touching the head? What is the difference between Left and Right Sham, then? This should be explained better and Fig 4 should depict the actual situation. Right now, it looks like the coil was held in the same exact location for the Sham and TMS conditions.

Is it possible that there are some spatial attention priming or adaptation effects? After hearing a clicking noise at 0.75 Hz for 5 minutes coming from the left or the right side (due to the coil being place on LM1 or RM1), is it possible that people are somewhat biased in the left/right choices they make just because of the lateralized sensory inputs during the stimulation phase? I cannot think of specific studies showing this, but it’s also not my field. Note that the Sham conditions cannot disambiguate this as the Left and Right conditions were not really lateralized.

Was the order of conditions counterbalanced? How? Were the same stimuli associated with the same conditions for all subjects? The effects are rather large, and so the usual questions about order counfounds typical in these kinds of blocked TMS studies etc come into play. 

Line 131. What does it mean that subjects were unaware of what condition they were experiencing? Did you ask them? Obviously, they were aware that the stimulation phase was on the left or on the right.

Line 132. What does it mean that “subjects were not aware that images were both repeated and displayed on both screen sides”? People are good at detecting repetitions. Furthermore, did you ask them?

Results

Line 173+ If you do a 2x2 Chi Square test using ShamL, ShamR, ChoiceL, ChoiceR, that comes out significant, which is a bit puzzling. This should be reported and discussed.

Figure 4. There should be a panel showing the individual percentages.

It is odd that there are no effects on RTs, given that M1 is being stimulated, but there are high-level effects on choice side.

Figure 6. If there were 140 trials per subject, how come there are some people with more than 140 descriptors? Is it because some responses include more than one descriptor and these are counted multiple times? If you look at the No Descriptor dots in Fig 6A, they are all under 20, and yet, the columns in panel B for the Not Sure category are all above 20. This is confusing, and so the legend should be improved to explain things, assuming there isn’t an issue with the numbers themselves. Also, it is odd to have a table inside the figure, unless it is labelled as a separate panel.

Lines 184+ This sentence is difficult to understand.

Lines 211+ this part is difficult to understand because of the earlier confusion about stimulus numbers and whether the stimuli are repeated across conditions (and how many times). For the t-tests, would be good to provide Cohen’s d so that people can use the data in future meta-analyses. Also, calculating Bayes’ factor would provide additional information.

Discussion.

The discussion should try to go a little deeper into what may be going on here. Why would one expect inhibition of the left motor cortex to lead to a preference for left side stimuli? Why not the other way around? 

The English is generally fine. 

Author Response

Comments and Suggestions for Authors

This is a very intriguing study on the effects of low 0.75 Hz rTMS to the left or right M1 on left/right stimulus choices (made with the left or right hand). The results show that left M1 inhibition biases people to choose the left side image with the left hand (and vice versa for right M1 inhibition). Furthermore, people seem to rationalize their choices, indicating they are not really aware of what drives them.

We thank the reviewer for the comments. We agree the study is intriguing, and we appreciate the positive feedback. We added/took all of the comments into the paper. The paper is now improved. In the very rare cases that a comment was not fully incorporated, the reason is explained and in all cases we tried to add the change and saw that the result was not optimal. That said, we now think the paper to be much improved and we cannot thank the reviewer enough for their help and assistance.

I have a number of comments and questions about the paper. Overall, there needs to be clarification on key aspects of the methodology.

---

Introduction. The predictions should spell out the direction of effects: left M1 stimulation is expected to …

In the last paragraph off the intro, the predicted direct effect of Motor Cortex stimulation has been added, indicated that inhibition to the Left MC would lead to more left-side responses and vice versa.

Stimuli. There is mention of 140 instances of a choice. Does this mean that there were 140 distinct pairs of picture people could choose from? Or were some pictures/pairs repeated? Why does the Experiment section say that “participants were presented with 60 stimuli consisting of two similar pictures and instructed to choose between the two and then explain why” (Line 154)? Weren’t participants presented with 140 stimuli?

This error was corrected within the text and should say “participants were presented with 70 stimuli…”

Experiment. Given that Sham was at Cz whereas stimulation of the motor cortex was lateralized, the coil sound was also lateralized. For the sham it is said that the coil was held at 90 decgrees over Cz. What does that mean? That the edge of the coil was touching the head? What is the difference between Left and Right Sham, then? This should be explained better and Fig 4 should depict the actual situation. Right now, it looks like the coil was held in the same exact location for the Sham and TMS conditions.

We have changed this and clarified both in the text and in the figure

Is it possible that there are some spatial attention priming or adaptation effects? After hearing a clicking noise at 0.75 Hz for 5 minutes coming from the left or the right side (due to the coil being place on LM1 or RM1), is it possible that people are somewhat biased in the left/right choices they make just because of the lateralized sensory inputs during the stimulation phase? I cannot think of specific studies showing this, but it’s also not my field. Note that the Sham conditions cannot disambiguate this as the Left and Right conditions were not really lateralized.

We have added this possibility to the discussion.

Was the order of conditions counterbalanced? How? Were the same stimuli associated with the same conditions for all subjects? The effects are rather large, and so the usual questions about order confounds typical in these kinds of blocked TMS studies etc come into play.

This has been added and explained.

Line 131. What does it mean that subjects were unaware of what condition they were experiencing? Did you ask them? Obviously, they were aware that the stimulation phase was on the left or on the right.

We went back in to clarify that subjects were unaware if experiencing inhibition or Sham to either the left or right motor cortex.

Results

Line 173+ If you do a 2x2 Chi Square test using ShamL, ShamR, ChoiceL, ChoiceR, that comes out significant, which is a bit puzzling. This should be reported and discussed.

As a control, we compared the two sham conditions in terms of Screen Side Responses and performed a Bonferroni correction. We have now added and clarified this.

Figure 4. There should be a panel showing the individual percentages.

We spent a large amount of time trying to do this. It did not look right (we tried scatter, pie, and bar) and only works as its own figure.  Even here, because presenting left and right itself is redundant (it always adds to 100%, the figure was not helpful). We could add this, but winds up being redundant-two figures showing the same thing.

It is odd that there are no effects on RTs, given that M1 is being stimulated, but there are high-level effects on choice side.

Agreed. We were surprised but think that variability played a role within each condition.  If one looks at the reaction time differences, there is huge variability across the subjects.

Figure 6. If there were 140 trials per subject, how come there are some people with more than 140 descriptors? Is it because some responses include more than one descriptor and these are counted multiple times? If you look at the No Descriptor dots in Fig 6A, they are all under 20, and yet, the columns in panel B for the Not Sure category are all above 20. This is confusing, and so the legend should be improved to explain things, assuming there isn’t an issue with the numbers themselves. Also, it is odd to have a table inside the figure, unless it is labelled as a separate panel.

We have added that multiple descriptors were used for a single stimulus. The table was meant to be separate and we have added that. I think the formatting of the paper dropped our legend. It is now listed as Table 1.

Lines 184+ This sentence is difficult to understand.

These sentences were edited to clarify Screen Side Responses based on which Sham condition was employed.

Lines 211+ this part is difficult to understand because of the earlier confusion about stimulus numbers and whether the stimuli are repeated across conditions (and how many times). For the t-tests, would be good to provide Cohen’s d so that people can use the data in future meta-analyses. Also, calculating Bayes’ factor would provide additional information.

We have added Cohen’s D to emphasize the effect size. We agree that we should have added this. Our apriori was a pure guess so we left out Bayes. However, now that we have data on this task, we have established the ability to do Bayes in the next studies. Other reviewers encouraged follow-up studies which we will do.

Discussion.

The discussion should try to go a little deeper into what may be going on here. Why would one expect inhibition of the left motor cortex to lead to a preference for left side stimuli? Why not the other way around?

We have added this. This was clarified in part in the Methods as well.

Reviewer 2 Report

Line #133

The study investigates the role of extraneous stimulation (in this case TMS), on conscious decision making process, with low frequency TMS, intending to temporarily inhibit the brain activity in selected locations.  They report that the TMS stimulation on either hemispheric region can have an impact on the conscious decision.

Overall, this study demonstrates some interesting possibilities of extraneous stimulation in exploring the working of consciousness.

I have the following concerns/ comments

#1: The method section suggests that the TMS is expected to disrupt the neurons in motor cortex. Motor cortex is directly related to the performance of the movement.  If the authors were interested in disrupting the decision-making process, wouldn’t it be better to stimulate the premotor or even prefrontal cortex?  Was there any reason to stimulate the motor cortex directly?

#2: The report indicates that there was no significant changes  (lines 162- 168)in the reaction time between sham and stimulation category. I wonder if the reaction time could have been changed by the actual disruption of the circuit activity, which is not seen in this study.   It would be nice if the authors gave details on that and comment from a neurophysiological standpoint.

Minor comments :

 Line # 184: 0.75 Hz

Moderate level of editing suggested. 

Author Response

The study investigates the role of extraneous stimulation (in this case TMS), on conscious decision making process, with low frequency TMS, intending to temporarily inhibit the brain activity in selected locations.  They report that the TMS stimulation on either hemispheric region can have an impact on the conscious decision.

Overall, this study demonstrates some interesting possibilities of extraneous stimulation in exploring the working of consciousness.

We appreciate the positive feedback and we thank the reviewer for taking the comment to improve the manuscript.

I have the following concerns/ comments

#1: The method section suggests that the TMS is expected to disrupt the neurons in motor cortex. Motor cortex is directly related to the performance of the movement.  If the authors were interested in disrupting the decision-making process, wouldn’t it be better to stimulate the premotor or even prefrontal cortex?  Was there any reason to stimulate the motor cortex directly?

This is an excellent insight that we debated and actually wanted to expand the study to include regions from the fronto-polar to SMA to M1. Obviously, the study would have been beyond practical scope and we limited ourselves to 4 regions (2 active/2 sham). The reason M1 was chosen was it is the one area we can clearly know that we have stimulated/located without the aid of neuronavigation and we know the same regions was targeted across all subjects. Because we know M1 can influence decision making (Klein JON 2012; Tecilla BrSci 2022) we chose this area. We are planning on doing what is suggested here now that we know we can get results with this paradigm. We are most interested now in Bode’s work and want to pursue those regions. This work, however, showed different regions across studies (hemispheric regions ‘flipped’) which made us concerned about where to stimulate-especially for a first study. That said, this current study is critical in establishing that the motor decision network can be manipulated as we hoped.

#2: The report indicates that there was no significant changes  (lines 162- 168)in the reaction time between sham and stimulation category. I wonder if the reaction time could have been changed by the actual disruption of the circuit activity, which is not seen in this study.   It would be nice if the authors gave details on that and comment from a neurophysiological standpoint.

Again, we don’t know until we do the follow up study outlined above. That is, we would need to stimulate multiple regions within participants across multiple trials. We have added this to the discussion because we agree with the reviewer that RT may be changed via SMA or FPC TMS. We think that RT variability between subjects played a role here, but the bigger point is that this study is the first of many planned studies.

Reviewer 3 Report

The study “Transcranial Magnetic Stimulation Alters Freewill Without Conscious Awareness” addresses an important and relatively unexplored area of research, adding to the existing body of knowledge on decision-making and the potential influence of neural processes. The study aims to shed light on the complex interplay between cognitive-motor functions and conscious choices, ultimately questioning the nature of free will. While the paper presents a thought-provoking inquiry into this area of neuroscience, there are some  aspects that require critical evaluation.

Title: It is very important to consider changing the title of the article. The use of a philosophical and ambiguous term like “free will” does not have a clear or agreed-upon definition in science. There is no hypothesis or research question that guides the study and explains how to measure the amount of free will and separate it from determinism.

I suggest the following alternatives:

  • “Effects of transcranial magnetic stimulation on neuronal activity and decision making”
  • “Alterations in brain activity induced by transcranial magnetic stimulation and their relation to decision making”

Interpretation of Results: While the study highlights interesting findings related to screen-side preferences, it could benefit from a more in-depth discussion of the implications of these findings and their broader relevance to the field of neuroscience and psychology. Moreover, the article does not sufficiently address the implications and limitations of the results. It does not explore the possible explanations for the observed effects nor acknowledge the potential confounding variables that could have influenced the results.

The article could be significantly enhanced by a more in-depth discussion that explores the potential causes behind the effects of TMS on the selection of behavior. This includes examining the specific elements that are impacted to create a bias in behavioral responses when using TMS. Furthermore, it would be beneficial to delve into why these effects are observed in a certain percentage of cases, rather than being an all-or-nothing phenomenon. Given the impressive results obtained from the study, such a comprehensive discussion is warranted and would provide valuable insights into the findings.”

Minor point. Writing Style: while generally clear, the writing style could benefit from improved flow and readability in certain sections. Some sentences are lengthy and complex, which may make it challenging for readers to follow the arguments.

Author Response

The study “Transcranial Magnetic Stimulation Alters Freewill Without Conscious Awareness” addresses an important and relatively unexplored area of research, adding to the existing body of knowledge on decision-making and the potential influence of neural processes. The study aims to shed light on the complex interplay between cognitive-motor functions and conscious choices, ultimately questioning the nature of free will. While the paper presents a thought-provoking inquiry into this area of neuroscience, there are some  aspects that require critical evaluation.

Title: It is very important to consider changing the title of the article. The use of a philosophical and ambiguous term like “free will” does not have a clear or agreed-upon definition in science. There is no hypothesis or research question that guides the study and explains how to measure the amount of free will and separate it from determinism.

I suggest the following alternatives:

  • “Effects of transcranial magnetic stimulation on neuronal activity and decision making”
  • “Alterations in brain activity induced by transcranial magnetic stimulation and their relation to decision making”
  • The title of the paper has been changed to reflect decision-making processes instead of a focus on the long-debated term of ‘free will’.

Interpretation of Results: While the study highlights interesting findings related to screen-side preferences, it could benefit from a more in-depth discussion of the implications of these findings and their broader relevance to the field of neuroscience and psychology. Moreover, the article does not sufficiently address the implications and limitations of the results. It does not explore the possible explanations for the observed effects nor acknowledge the potential confounding variables that could have influenced the results.

We agree we could have discussed this in more depth. We have added these requested comments. We thank the reviewer for pointing this out and clearly the paper is greatly improved.  We now speculate without going beyond our data.

The article could be significantly enhanced by a more in-depth discussion that explores the potential causes behind the effects of TMS on the selection of behavior. This includes examining the specific elements that are impacted to create a bias in behavioral responses when using TMS. Furthermore, it would be beneficial to delve into why these effects are observed in a certain percentage of cases, rather than being an all-or-nothing phenomenon. Given the impressive results obtained from the study, such a comprehensive discussion is warranted and would provide valuable insights into the findings.”

We have added this and the discussion is now expanded. We think this point is important and we thank the reviewer there. This should have been thought of and we missed it (the all-or-nothing)

Minor point. Writing Style: while generally clear, the writing style could benefit from improved flow and readability in certain sections. Some sentences are lengthy and complex, which may make it challenging for readers to follow the arguments.

We shored up many of the sentences.

Reviewer 4 Report

Authors of this manuscript investigated how inhibiting the motor cortex by using Transcranial Magnetic Stimulation (TMS) influences cognitive-motor functions involved in decision-making. They found that inhibitory TMS to either the left or right motor cortex led to a significant bias in screen side preference during the choice assays, suggesting that TMS can unconsciously affect the cognitive-motor processes behind decision-making. However, the time to make a decision and their explanation for their choices cannot be influenced by TMS. This study can help us understand the intricate relationship between neural circuits and conscious intention. However, manuscript can be improved much to make it better. I have some comments about the manuscript as followings.

Major comments: 

1. One major concern about the experiment design is without control. All data involved in testing decision-making are from experimental groups without control group. How do authors think experimental groups only can lead them to make a conclusion? Or do authors have some reasons why they don’t have control ? 

2. Authors used TMS of 0.75 Hz, does 0.75 Hz have any specialty for study like this? will different frequency cause different effects about the decision-making? Will stronger stimulation have more stronger influence? 

3. About participants in Materials and Methods, 11 females and 6 males were recruited for the study. Is the number of participants too small? Will this sex difference in number affect the results? Additionally, what is the ages of these participants?

4. In line 97, does the difference itself between left-handed and right-handed participants have effects on the experimental results given that participants were asked to choose the images using their left or right hands?

5. Line 115, 5 seconds were given to participants to explain why they chose the image. Does this time duration matter much?

6. For Figure 2, I think the arrangement is not so clear to read. It is difficult to easily see what is included in A is and what is included in B. Making the figure clear will help a lot. Rearrangement is needed. 

7. About the reaction time of Figure 3, how is the reaction time determined ? what is method to calculate the time? It should be written clearly in Methods.

8. In Figure 5, for Right 0.75 TMS and Left sham TMS, why authors don’t put the texts “feeling” into the blue just like other two ?

Minor comments:

1. line 47, “…the participant themselves is a…”, “is” should be “are”?

2. line 121, “Participants were read the informed consent…”, does it need correction? 

Author Response

Authors of this manuscript investigated how inhibiting the motor cortex by using Transcranial Magnetic Stimulation (TMS) influences cognitive-motor functions involved in decision-making. They found that inhibitory TMS to either the left or right motor cortex led to a significant bias in screen side preference during the choice assays, suggesting that TMS can unconsciously affect the cognitive-motor processes behind decision-making. However, the time to make a decision and their explanation for their choices cannot be influenced by TMS. This study can help us understand the intricate relationship between neural circuits and conscious intention. However, manuscript can be improved much to make it better. I have some comments about the manuscript as followings.

We thank the reviewer for the comments. We agree the study is intriguing, and we appreciate the positive feedback. We added/took all of the comments into the paper. The paper is now improved. In the very rare cases that a comment was not fully incorporated, the reason is explained and in all cases we tried to add the change and saw that the result was not optimal. That said, we now think the paper to be much improved and we cannot thank the reviewer enough for their help and assistance.

Major comments:

  1. One major concern about the experiment design is without control. All data involved in testing decision-making are from experimental groups without control group. How do authors think experimental groups only can lead them to make a conclusion? Or do authors have some reasons why they don’t have control ?

Sham is our control. Further, we did not use one hemisphere, we used BOTH. That is, we found the right AND the left had effects to the predicted direction. We find this as very strong experimental evidence. True, we used one task, but with Sham and multiple experimental conditions we are confident in our design. In other words, the right TMS caused the opposite effects compared to the left TMS and the sham had no effect.

We are aware that any experiment can be imporved and certainly this is no exception. You will read that this experiment leads to more questions than it answers. Like the reviewer, we too would like to know about handedness and sex differences. Further, we would like to know about different stimuli and different TMS levels.

  1. Authors used TMS of 0.75 Hz, does 0.75 Hz have any specialty for study like this? will different frequency cause different effects about the decision-making? Will stronger stimulation have more stronger influence?

These were IRB limited parameters. We have been using this/these parameters since 2001 when the laboratory was established. We assumed that 10Hz TMS would give us opposite effects. We also assume that stronger TMS would give the same results in terms of nature, but larger effect sizes. Because we do not have a medical school or physician on staff, we are very safety conscious. As a non-R1, we have published our success in running a safe, effective lab that uses TMS elsewhere.  Shelanskey, T.,  Chavarria, K., Pagano, K., Sierra, S., Martinez, V., Ahmad, N., Brenya, J., Janowska, A., Zorns, S., Straus, A., Mistretta, V., Balugas. B., Pardillo, M., & Keenan, J. P. (2022). Employing Transcranial Magnetic Stimulation in a Resources Limited Environement to Establish Brain Behavior Relations. Journal of Visualized Experiments.         

  1. About participants in Materials and Methods, 11 females and 6 males were recruited for the study. Is the number of participants too small? Will this sex difference in number affect the results? Additionally, what is the ages of these participants?

We have added the ages. We have found reliable, replicated, validated results with 12-16 particiapnts in many of our studies. We have no idea if sex plays a role in this task. We would need to perform an additional study which we may do in the future. Duran, K., O'Halloran, H., Soder, H., Yasin, S., Kramer, R., Rosen, S., Brenya, J., Chavarria, K., Savitska, L., & Keenan, J. P. (2020). The Medial Prefrontal Cortex: A potential link between self-deception and affect. International Journal of Neuroscience.

Kramer, R., Duran, K., Soder, H., Applegate, L., Youssef, A., Criscione, M., & Keenan, J. (2020). The Special Brain: Subclinical Grandiose Narcissism and Self-Face Recognition in the Right Prefrontal Cortex. The American Journal of Psychology, 133(4), 487-500. doi:10.5406/amerjpsyc.133.4.0487

  1. In line 97, does the difference itself between left-handed and right-handed participants have effects on the experimental results given that participants were asked to choose the images using their left or right hands?

Similar to the next response, this we do not know. We have tried experiments recruiting LH participants and it took us 200 people to get 12 pure LH participants at our institution. Perhaps we can do a follow-up study.

  1. Line 115, 5 seconds were given to participants to explain why they chose the image. Does this time duration matter much?

We do not know this. We chose this based on a few pilot subjects where it seemed long enough to get a response.

  1. For Figure 2, I think the arrangement is not so clear to read. It is difficult to easily see what is included in A is and what is included in B. Making the figure clear will help a lot. Rearrangement is needed.

We have simplified and changed it

  1. About the reaction time of Figure 3, how is the reaction time determined ? what is method to calculate the time? It should be written clearly in Methods.

We have clarified this in the text. We should have done this initially.

  1. In Figure 5, for Right 0.75 TMS and Left sham TMS, why authors don’t put the texts “feeling” into the blue just like other two ?

It did not fit without changing the font which looked a lot worse.

Minor comments:

  1. line 47, “…the participant themselves is a…”, “is” should be “are”?

We have made this change in the text.

  1. line 121, “Participants were read the informed consent…”, does it need correction?

This sentence was clarified in the manuscript.

Round 2

Reviewer 4 Report

Authors addressed my comments. I agree now it can be accepted for publication.